# Fracture Mechanisms of S355 Steel—Experimental Research, FEM Simulation and SEM Observation

**DOI:** 10.3390/ma12233959

**Published:** 2019-11-29

**Authors:** Ihor Dzioba, Sebastian Lipiec

**Affiliations:** Department of Machine Design, Faculty of Mechatronics and Mechanical Engineering, Kielce University of Technology, Al. 1000-lecia PP 7, 25-314 Kielce, Poland; slipiec@tu.kielce.pl

**Keywords:** S355 ferritic steel, strength properties, fracture toughness, fracture mechanisms

## Abstract

In this study, the fracture mechanisms of S355 ferritic steel were analyzed. In order to obtain different mechanisms of fracture (completely brittle, mixed brittle and ductile or completely ductile), tests were carried out over a temperature range of −120 to +20 °C. Our experimental research was supplemented with scanning electron microscopy (SEM) observations of the specimens’ fracture surfaces. Modeling and load simulations of specimens were performed using the finite element method (FEM) in the ABAQUS program, and accurate calibration of the true stress–strain material dependence was made. In addition, the development of mechanical fields before the crack tip of the cracking process in the steel was analyzed. The distributions of stresses and strains in the local area before the crack front were determined for specimens fractured according to different mechanisms. Finally, the conditions and characteristic values of stresses and strains which caused different mechanisms of fracture—fully brittle, mixed brittle and ductile or fully ductile—were determined.

## 1. Introduction

Structural elements made of ferritic steel are commonly used in various types of structures and mechanisms over a wide range of temperatures. In order to design them correctly and ensure their safe operation, it is essential to provide information on strength characteristics and fracture toughness of the material within the range of service temperatures. Modern technologies of production and inspection do not allow the presence of crack-type defects in newly manufactured components. However, during long-term operation in conditions of cyclic loads and environmental impact, cracks in the elements may arise and develop from microstructural defects (e.g., from particles of large inclusions) [1,2]. A particularly high risk of initiation and development of cracks in a component is found when inclusions are grouped in one plane, which leads to the development of internal delamination cracks [3,4,5,6]. Defects in the form of cracks often occur in welded joints as well [7,8]. In this case, crack initiation is possible in the joint material or in the heat-affected zone.

Methods for estimating the strength of elements containing crack-type defects are presented in the FITNET procedures [9,10,11]. These procedures are based on knowledge of standard characteristics of strength (σ*_YS_*, σ*_UTS_*) and fracture toughness (*K*_IC_, *J*_IC_, δ_TC_), leading to relatively high conservatism in the results obtained when assessing the strength of a structural element. A conservative assessment can be reduced by performing the analysis at a higher level of advancement, taking into account the influence of in-plane and out-of-plane constraint on the fracture toughness characteristics.

In this work, the critical state of the material at moment initiation of brittle or ductile cracking was evaluated by analyzing the distribution of stresses and strains in the local zone in front of the crack tip where the material was most stressed. Conceptions concerning analysis of the fracture process in the local zone were proposed in the middle of the 20th century [12,13,14,15,16,17,18]. However, wide application of the local approach to fracture (LAF) and strength analyses was only possible with the development of numerical computational methods and computer technology. The LAF methodology requires detailed knowledge of the distributions of stress and strain in the local zone and determination of their critical values (σ_C_, *ε*_C_). Three basic stages are distinguished in the fracture process: (i) nucleation of microdefects; (ii) development of microdefects within the microstructural unit (e.g., grains and post-martensite laths); and (iii) propagation of microdefects across boundaries into adjacent grains—i.e., the formation of a microcrack [19,20,21]. Depending on the microstructure of the material and the stress–strain state in the analyzed zone, fractures may propagate by a brittle or ductile mechanism. Taking into account the heterogeneity of the microstructure of ferritic steels, scattering of critical characteristics is observed as well. Based on the Weibull distribution [22], Beremin proposed a statistical description of the fracture process [23].

Temperature has a major influence on the nature (brittle or ductile) of fracture development. Depending on the temperature, different fracture mechanisms can occur within the same material: completely brittle, mixed brittle and ductile or completely ductile. The problems associated with fracture at different temperatures have been given a great deal of attention. The statistical fracture model proposed by Beremin has been adapted to describe the fracture process using critical fracture toughness values (*K*_IC_, *K*_JC_) for brittle and brittle–plastic fractures, and is presented as dependence *K*_JC_ = *f*(*T*), commonly known as the Master Curve [24,25,26]. Following some simplifications and the selection of coefficients, it is possible to use Master Curves to predict brittle fracture in ferritic steels, for which 275 ≤ σ_YS_ ≤ 825 (MPa). When assessing the fracture toughness in steels with a higher yield strength, σ_YS_ > 825 (MPa), and, for *K*_JC_ = *f*(*T*), the appropriate coefficients must be determined [27,28,29].

Studying brittle fractures, Ritchie, Knott and Rice (RKR) concluded that brittle fractures occur if the normal component of stresses to the plane of a crack exceeds a critical level (σ*_ii_* > σ_C_) on a section longer than a critical length (*l* > *l*_C_) [18]. In the studies of Neimitz et al., the RKR criterion was modified by the introduction of a model of large finite strains during the calculation of mechanical fields in front of the crack tip, which made it possible to determine the exact shape of their distributions in this zone [30,31]. Despite the modification, the problem of determining the critical stress level and critical section length for a specific steel grade remains.

According to the basic model of crack growth by a ductile mechanism, crack propagation is a multistage process that includes the nucleation, growth and coalescence of voids. Through modeling of void initiation, presented in such papers as Refs. [32,33,34,35], a number of models and formulas for estimating the critical values of stresses and strains have been proposed, which ultimately result in inclusion cracking or decohesion between a particle and the ferritic matrix. In a description of void growth by Rice and Tracey [17], void growth rate is said to depend on characteristics of the stress field and plastic strains present in the material around the voids. In addition, damage by coalescence of voids and the formation of pre-existing fractures have been described by means of models based on the Gurson, Needleman, Tvergaard (GNT) coherence model [36,37].

This paper presents research results obtained from specimens of S355 steel. Experiments were carried out over temperatures ranging from −120 to 20 °C on uniaxially stretched cylindrical specimens and notched beam-shaped specimens, which were bent according to a three-point scheme (the single-edge notched bend [SENB] scheme). Through the selection of a wide range of temperatures, it was possible to obtain different mechanisms of fracture for the SENB specimens: brittle, brittle–ductile and ductile. In order to precisely identify the fracture mechanism, experimental tests were supplemented with fractographic examination of the surface of breaks using scanning electron microscopy (SEM).

Numerical modeling and finite element method (FEM) simulation of the loading process using the ABAQUS software were performed on selected specimens with characteristic fracture mechanisms. As a result, stress and strain distributions in the local zone in front of the crack tip were possible to obtain. Then, critical and characteristic values of stress and strain were determined for the damage-causing mechanisms by the comparison of the characteristics of stress and strain fields with the results of SEM observations.

Some research results were previously partially presented and published by the authors [38,39,40,41,42]. The current version includes extended and supplemented research results.

## 2. Materials, Research Methods and Characteristics of the Materials

The tests were conducted on specimens made of S355 steel (formerly known as 18G2A), which is widely used in various types of structures. The chemical composition of S355 steel in weight % is [43]: C-max 0.24; Si-max 0.55; Mn-max 1.60; P-max 0.035; S-max 0.035; N-max 0.012; and Cu-max 0.55. In its original state, S355 steel has a layered ferritic–pearlitic microstructure (Figure 1a), resulting in significant anisotropy of mechanical characteristics depending on the direction of cutting. In order to homogenize the state of the microstructure (and mechanical characteristics, accordingly), the specimens were subjected to laboratory heat treatment before the final mechanical processing. This involved hardening in oil at 950 °C, annealing at 600 °C for 150 h, and cooling in water. As a result, we obtained a homogeneous microstructure of ferrite with coagulated particles of precipitates along the grain boundaries (Figure 1b), with the size of ferrite grains ranging from 2 to 12 µm and carbide particles from 0.1 to 0.8 µm. 

The stress–strain dependences of S355 steel in nominal and true values, as well as the strength and plasticity characteristics, were determined by measuring the force and elongation of the extensometer as registered during the uniaxial tensile tests. The tests were carried out in accordance with ASTM recommendations [44] and five cylindrical specimens with diameters of 5.0 mm were used. Fracture toughness characteristics were determined for the SENB specimens using the ASTM method, which is based on method of compliance changes in specimen during crack growth [45]. The tests performed in negative temperature conditions were carried out in a thermal chamber in an environment of nitrogen evaporation, maintaining the temperature within ±1.0 °C. Experimental tests were carried out on the universal testing machine ZWICK-100, equipped with automated control and data recording systems. In order to identify the mechanisms of fracture development, fractographic examinations of the fracture surface of selected specimens were carried out using SEM. Cross-sections of the metallographic specimens, which were perpendicular to the main direction of the subcritical crack, were also studied. This allowed us to obtain additional qualitative confirmation of the nature of the fracture. Based on numerical modeling of the specimens and simulations of their loading using ABAQUS software, stress and strain fields were obtained in the most stress-affected local zone—in front of the crack tip.

In order to establish the characteristic mechanisms of subcritical crack development, tests on the SENB specimens (*B* = 12.0; *W* = 24; *S* = 96 mm; *a*_0_/*W* = 0.5) were carried out in temperatures ranging from −120 to 20 °C. The dependence of the J-integral (*J*_IC_ or *J*_C_) critical values on temperature was obtained, as shown in Figure 2a. There were three characteristic intervals in the temperature dependence of the critical values of fracture toughness: (i) the area of the low plateau, for *T* ≤ −100 °C, where a completely brittle fracture by cleavage of the subcritical crack was observed (Figure 2b); (ii) the zone of mixed fracture, −100 °C < *T* < −50 °C, where the crack grew by both brittle-cleavage and ductile mechanisms, with the latter being through the development of voids (Figure 2c); and (iii) the area of the upper plateau, *T* ≥ −50 °C, where crack growth was completely ductile (Figure 2d). As the temperature increases, the plasticity of the material increases. An increase in the participation of ductile growth mechanisms during propagation of the subcritical crack and an increase in the critical value of *J*_IC_, as well as the stretch zone width (SZW).

Uniaxial tensile tests of the cylindrical specimens were carried out at the temperatures specified above. On the bases of these tests, stress–strain dependencies and values of basic material characteristics were determined. In general, an increase in temperature caused a decrease in strength characteristics and an increase in plasticity of the material (Figure 3a,b). The nominal and true values of strength characteristics and critical values of fracture toughness for the S355 steel specimens tested over the temperatures specified are presented in the Table 1.

## 3. FEM Calculation: Specimen Modeling and Material Calibration

Numerical calculations were performed on the basis of a model of a three-point bent SENB specimen made in the ABAQUS program. The geometry of the specimen in the numerical model was the same as the specimens used in the experimental research (thickness *B* = 12 mm, high *W* = 24 mm, *a*_0_/*W* ≈ 0.55−0.6). Due to symmetry, only one-quarter of the SENB specimen was modeled. The following boundary conditions were defined for the numerical calculations (Figure 4): displacement of the cracked specimen plane *XOZ* is possible; displacement of the non-cracked part of plane *XOZ* of the specimen in the *y* direction is blocked; displacement of the specimen’s middle plane *XOY* in the *z* direction is blocked; and the supporting roll is completely immobilized. In the numerical computations, displacement of the roll loading the specimen was forced. In the model, 8-node, three-dimensional finite elements were used. The SENB specimen in the model was divided into 21 layers in the direction of thickness. The size of the finite elements used and the width of the layers were selected after taking into account the condition of convergence of the results obtained. The size of the finite element decreases when approaching the crack front. The front (top) of the crack was modeled as an arc with a radius of 0.012 mm.

For the numerical calculations, load was introduced by forcing the displacement (from pressure) of the upper roll on the surface of the SENB specimen. Numerical simulations of loading were carried out according to the true load recorded during the experimental tests. Numerical calculations were made for different load stages (Pi), wherein the results obtained from the extensometer indicated characteristic fracture types: brittle, brittle-ductile and ductile (Figure 5).

In order to perform numerical calculations, especially of elements in materials with high levels of plasticity, it is necessary to properly calibrate the constitutive stress–strain relationship of the material, as obtained from uniaxial tensile tests. Proper calibration of the material is of particular importance when analyzing mechanical fields in the immediate vicinity of a crack tip; a situation relevant to our considerations. Calibration of stress–strain dependence of a material is performed according to the method proposed by Bai and Wierzbicki [46,47], who were the first to suggest an approach that used the plasticity function, as well as by the method proposed by Neimitz et al. [40,41,42], who introduced the softening function to the calibration procedure. The softening phenomenon is associated with the process of coalescence of voids in the material just prior to violent destruction. The calibration procedure utilizes knowledge of numerically-determined distributions of effective stresses *σ*_eff_, mean stresses *σ*_m_, the stress triaxiality factor *η* = *σ*_m/_*σ*_eff_, effective plastic strain *ε*_eff_plast_ and the Lode parameter, *L*.

In the first step of the calibration procedure, the true stress–strain dependence, which was obtained based on data from the uniaxial tension tests, was extrapolated to a high strain level (≈250%). The extrapolation was based on the description of the linear law of endpoints from true elongation of the stress–strain dependence and took into account about 300 endpoints. Next, using the formulas described in Refs. [40,41,42], the best possible correlation between the numerically and experimentally obtained force–elongation curves was sought. As a result of the above-described calibration procedure of the constitutive relationship of the material, a very good correlation of the force–elongation curves was achieved.

## 4. Numerical Calculation Results

### 4.1. Brittle Fracture by Cleavage

Brittle cleavage fracture occurred (Figure 2b) in the ferritic steel specimens at temperatures corresponding to the lower plateau of the brittle-to-ductile transition dependence. This type of fracture usually takes place in negative temperature conditions, below −80 °C. However, in some heat-resistant ferritic steels operated for a long time in high temperatures (e.g., 14MoV6-3, 14MoV6-9, DIN Standard), an increase in the service time of the elements can result in a significant shift. In such steels, brittle fractures may be observed at higher temperatures, even above 0 °C [48].

According to the assumptions of the modified RKR criterion [18], brittle fractures will occur if the level of stresses normal to the fracture plane exceeds the critical level (*σ_ii_* > *σ*_C_) over a distance longer than the critical length (*l* > *l*_C_). In accordance with the loading scheme and coordinate system adopted in the model, the direction of stress was normal in relation to the crack plane (*σ*_22_). Completely brittle fractures were obtained in the specimens tested at temperatures below −100 °C. Numerical calculations were carried out for a specimen tested at *T* = −120 °C. Distributions of the stress (*σ*_11_—stress in the crack’s propagation direction, *σ*_22_—normal stress to the crack plane, *σ*_33_—stress in the specimen’s thickness direction) and strain in the middle layer of the SENB specimen in the zone in front of the crack tip are shown in Figure 6a. The maximum values obtained for strains in these conditions reached 25–30% immediately before of the crack tip and rapidly decreased as distance from the crack tip increased. At a distance of 0.1 mm from the crack front, strains were less than 2%. As the distance from the crack tip increased, the stress levels increased, reaching their maximum values before decreasing gradually. The highest values in the analyzed zone were reached by *σ*_22_, the normal stress to the crack plane (*σ*_22_max_ = 1760 MPa, *σ*_22_max/_*σ*_YS_T_ = 3.2). Since they have a leading role in the brittle fracture process, we analyzed stress distributions in more detail.

The stress *σ*_22_ distributions in parallel to the middle layers, from central to lateral planes, are presented in Figure 6b. The change of stresses *σ*_22_ in the thickness direction at some distance from the crack tip are shown in Figure 6c. Based on the data presented in Figure 6b,c, one can see that the stresses *σ*_22_ possess high values at a certain width and distance from the crack tip, but fall as they move away from the front of the crack, decreasing quickly at the lateral planes of the specimen.

As brittle cleavage occurs directly at the tip of the fatigue pre-crack (Figure 6a), where the stresses *σ*_22_ ≈ 1550–1570 MPa, a stress value in this range was assumed as the critical stress level: *σ*_C_ = 1550 MPa. On the basis of the calculations performed, it was established that stress values of *σ*_22_ which exceeded the critical level *σ*_C_ (*σ*_22_ ≥ *σ*_C_ = 1550 MPa) occurred in a limited area before the front of the crack tip. For the tested SENB specimen, the shape and dimensions of this zone (*A*_1_) are shown in Figure 8d. The maximum length values of the area were found in the middle layers of the specimen, ranging between 340 and 360 µm at a width ± 2.5 mm from the middle plane, after which the length of the area decreased. At a width of ± 5.0 mm, the area disappeared. Base on the results obtained, it can be assumed that the dimension of the critical area was *A*_1_ ≈ 2.88 mm^2^, and the maximum length of the critical section (in the middle plane), where *σσ*_22_ ≥ *σ*_C_ = 1550 MPa, was equal to ∆*l*_C_ ≈ 350 µm.

### 4.2. Mixed Type of Fractures—Cleavage and Ductile Fractures

A fracture of mixed nature (ductile and brittle types) occurred in the transitional zone of temperature-dependent brittle-to-ductile dependence. At the breaks of the tested specimens in this range, there were two characteristic zones: ductile fracture and cleavage fracture (Figure 7). The ductile fracture zone was located directly at the pre-crack tip, after which was the area of the cleavage fracture. The width of the ductile fracture zone increased with an increase in temperature. As shown in Figure 7a, at *T* = −100 °C, the width of the plastic fracture zone was 12–15 µm and SZW was 20–22 µm. Then, at *T* = −80 °C, the width of the ductile fracture zone increased to ≈200 µm and SZW also increased to 75–85 µm.

In order to explain this type of fracture, the stress and strain fields of the SENB specimen tested at *T* = −80 °C were calculated. The calculations were conducted for several moments of specimen loading (Pi), allowing us to trace the development of the mechanical fields in the loading process (Figure 8). According to the stress diagrams, following an increase in the load, the maximum value of stress *σ*_22_ increased and moved away from the pre-crack tip (Figure 8a). At the moment when brittle fracture occurred (P5 in Figure 8a), some characteristic values were recorded: *σ*_22_max_ = 1676 MPa for *l*_max_ = 400 µm, and *σ*_22_max_/*σ*_YS_T_ = 3.3. The maximum length of the section, where *σ*_22_ ≥ *σ*_C_ = 1550 MPa, was equal to about ∆*l*_C_ ≈ 550 µm, from 210 to 760 µm. In the range where *σ*_22_ ≥ *σ*_C_, the strain level was low, *ε*_eff_plast_ < 20% (Figure 8b). However, the level of strain *ε*_eff_plast_ increased sharply, exceeding 50% over a segment of 100 µm immediately before the front of the crack tip, while at the tip it reached values ≈160%.

According to the stress distributions in the specimen thickness direction at the moment of loading P5, shown in Figure 8c, the area of stresses *σ*_22_ ≥ *σ*_C_ = 1550 MPa were located in zone *A*_2_, shown in Figure 8d. In the case of mixed cleavage and ductile fractures, the position and shape of the zone *σ*_22_ ≥ *σ*_C_ was significantly different from the one observed for completely brittle fractures (Figure 8d). The zone in this case was offset from the crack tip about 210 µm and it was longer (≈550 µm), although its width was smaller (≈6.4 mm). Despite this, the areas of the zones in both cases were similar: *A*_1_ ≈ 2.88 mm^2^ and *A*_2_ ≈ 2.76 mm^2^, and thus the lesser of these values can be accepted as a critical size, *A*_C_.

### 4.3. Ductile Fracture

A fully ductile fracture mechanism of SENB specimens of S355 steel was observed during the tests at *T* ≥ −50 °C (Figure 9). In these specimens, a subcritical fracture develops as a result of initiation, growth and coalescence of voids. In order to determine the conditions causing fracture development according to the ductile mechanism, numerical calculations were carried out for the mechanical fields of the SENB specimens, in which completely ductile fractures were recorded at *T* = −50 °C and *T* = 20 °C.

Qualitative character of stress and strain distributions at *T* = −50 and 20 °C is similar as in test at *T* = −80 °C (Figure 8a and Figure 10a,c). With loading increase the area of maximum values of stresses moves away from the crack tip. However, the increase in temperature causes a reduction in stress levels. In the central layer, the maximum normal stresses are: *σ*_22_max_ = 1600 MPa for *T* = −50 °C and *σ*_22_max_ = 1400 MPa for *T* = 20 °C. At *T* = −50 °C the stress level exceeds the critical level, *σ*_22_ ≥ *σ*_C_ = 1550 MPa, (maximum section length is–Δ*l* ≈ 400 µm and width is about 3.0 mm), on an area of *A*_3_ = 1.76 mm^2^. This area is smaller than the critical: *A*_3_ < *A*_C_, so the conditions for realization of the brittle fracture by cleavage are not met. During the test at *T* = 20 °C the stresses *σ*_22_ do not reach critical level, so brittle fracture cannot occur. While, the strain *ε*_e__ff_plast_ distributions for temperatures at which the ductile mechanism of crack growth occurred through initiation, growth and coalescence of voids are very similar (see Figure 7b and Figure 9b,d). In a section of up to 100 μm in front of the crack tip *ε*_eff_plast_ exceed 50% and immediately before the tip they reach up to 150%.

## 5. Discussion: Results of FEM and SEM Analysis

A summary of the results obtained are presented in Figure 11, Figure 12 and Figure 13 and in Table 2. All the results shown below were obtained for the local area before the pre-crack front and for layers in the central part of the specimen, presented as the plane strain state.

For the fully ductile mechanism of crack growth, the stress and strain distributions are presented in Figure 11a. The ductile mechanism is characteristic for cracking in the upper plateau, which for S355 steel is at *T* ≥ −50 °C. The stresses *σ*_22_ were found to be lower everywhere then the critical value of *σ*_C_ (Figure 11a); thus, brittle fracturing by cleavage could not occur. On the other hand, the strains *ε*_eff_plast_ immediately before crack the tip (in the section 0 to 0.2 mm away) reached high values, from 20% to 160%. SEM observations showed that for test temperatures *T* ≥ −50 °C, the length of the section on which the formation of voids takes place was similar, about ≈100 µm. Along this section, the level of plastic strain varies from 50% to 150%. Large voids developed and grew around particles of non-metallic inclusions, joining up with the main crack before the front of the crack tip (up to 25–30 µm), where the level of strain was 100%–160%, (Figure 11b). However, at a distance 30–100 µm from the crack tip, isolated large voids around non-metallic inclusions and minor voids around particles of carbide precipitates were mainly located along grain boundaries (Figure 11c,d). Thus, it can be stated that for initiation of the ductile fracture mechanism through the development of voids, a high level of plastic strain (above 50%) is necessary, while for crack growth a value over 100% is necessary.

Initiation of the cleavage fracture mechanism is achieved if, in the area before the crack tip, high stresses are present, namely normal stresses to the fracture plane (*σ*_22_ > *σ*_C_ = 1550 MPa), with a low level of plastic strain (*ε*_eff_plast_ < 0.20). Further, such a high level of stress should occur in a zone with an area greater than the critical *A**_C_*. In the case of the SENB specimens, with a net cross-section of 12 × 12 mm^2^, this area is about *A**_C_* ≈ 2.76 mm^2^. In the case of mixed fracture characteristics (Figure 12a), the *A**_C_* zone moved away from the crack tip by some distance, while in the zone between the pre-crack tip and *A**_C_* (where *σ*_22_ < *σ*_C_ and *ε*_eff_plast_ > 0.20), the crack growth was realized by a ductile fracture mechanism (Figure 12b). The shape of the *A*_C_ zone depends on its location, as it changes as it moves away from the front of the crack (Figure 8d). This is the result of stress field distributions in the specimen. Brittle fracture by cleavage was also not realized immediately, but cracking occurred first in separate grains at particles of carbide precipitates or inclusions (Figure 12c,d), followed by the connected of these microcracks with the main crack with an increase of loading.

For a completely brittle fracture by cleavage, the zone where *σ*_22_ > *σ*_C_ and *ε*_eff_plast_ < 0.20 was located directly before the front of the blunt pre-crack (Figure 13a). Some peculiarities of the realization of the cleavage fracture mechanism were determined on the basis of the observation of the fracture profile using SEM (Figure 13b). The cleavage fracture did not spread directly from the tip of the ductile crack or the fatigue pre-crack. At first, during the initiation stage, the separate micro-cracks were formed inside ferrite grains, arise along the slip planes (oriented about 45° to the loading direction). The size of these micro-cracks corresponded to the size of the grains, indicating that the size of the ferrite grains was one of the factors which determines the level of fracture toughness of the material, in addition to the strength properties.

The results obtained during comprehensive tests of the S355 steel fracture process in the range of brittle-to-ductile transition temperatures allowed for the establishment of certain characteristic rules, the fulfillment of which leads to the realization of an appropriate crack development mechanism. Models of ductile fracture development through nucleation, growth and coalescence of voids are based on the criterion proposed in Refs. [15,17], which show that the increase in voids depends on the level of plastic deformation and the stress triaxiality factor *η*:(1)$$\frac{dR}{R}≅0.283exp\left(1.5\eta \right)d\varepsilon_{eff\_plast}$$
where:(2)$$\eta =\frac{\sigma_{m}}{\sigma_{eff}}$$
(3)$$\sigma_{m}=\frac{\sigma_{11}+\sigma_{22}+\sigma_{33}}{3}$$
(4)$$\sigma_{eff}=\sqrt{3J_{2}}=\sqrt{3/2s_{ij}s_{ij}}=\frac{1}{6}\left[{\left(\sigma_{11}-\sigma_{22}\right)}^{2}+{\left(\sigma_{22}-\sigma_{33}\right)}^{2}+{\left(\sigma_{11}-\sigma_{33}\right)}^{2}\right]$$

So, the increase in voids depends on the level of stress triaxiality and an increase in the level of plastic deformation in the material:(5)$$\Delta R≅\Delta \varepsilon_{eff_{plast}}exp\left(\eta \right)$$

However, research shows that the relationship in Equation (5) describes the process of a void’s growth in certain ranges of *ε*_eff_plast_ and *η*. The development of a ductile fracture through nucleation, growth and joining of voids is possible when *ε*_eff_plast_ > 0.4, and the factor *η* is in the range of 0.7–1.0 (Table 2). When the SENB specimens were tested, this area was able to occur immediately before the top of the pre-crack. Similar ranges for sizes *ε*_eff_plast_ and *η* during ductile crack growth have also been obtained on specimens with other shapes [39,40].

On the other hand, brittle fracture by cleavage occurs at low levels of plastic deformation, *ε*_eff_plast_ < 0.2, higher levels of the stress triaxiality coefficient (1.5 < *η* < 2.4), and (correspondingly) at high levels of stress components (Table 2). These conditions must be met in an area with a specific surface, with its location and shape dependent on the type and geometric dimensions of the specimen. In the case that the conditions necessary for the realization of ductile and brittle mechanisms are met simultaneously during specimen loaded, we observe a mixed nature of crack propagation (Table 2).

The data of stress and strain fields which characterized possibility of realization the according type of fracture mechanism of cracking S355 ferritic steel are similar for one obtained at testing of AHSS Hardox-400 steel: for cleavage: *σ*_22_max_/*σ*_YS___T_ = 3.25; 1.5 < *η <* 2.34 and for ductile 0.2 < *ε*_eff_plast_ < 1.68 [49].

## 6. Conclusions

In this work, comprehensive research on the mechanisms of ferritic S355 steel fracture is presented. Tests were carried out in temperatures corresponding to the brittle-to-ductile fracture transition of the SENB-type specimens. Experimental research used as the basis for modeling and numerical simulations allowed for subsequent calculation and analysis of mechanical fields in areas before the top of the crack. Additional information regarding the development of fracture mechanisms was obtained by means of metallographic and fractographic studies, performed using SEM. Based on the results obtained during load simulation and numerical calculations, the nature of stress and strain distributions in the local area before the crack tip was determined.

The level of mechanical fields increased with increasing load. Effective plastic strains, *ε*_eff_plast_, increased monotonically when approaching the tip of the crack, and this increase was rapid in front of the pre-crack. The increase in load also caused a gradual increase in the value for the entire relationship.

The maximum value appears in the stress components diagram, the value of the maximum increased slightly and moved away from the vertex as the load increased.

The nature of the cracking was strictly dependent on the distributions of stress and strains in the local area, located in range ≈1.0 mm from the front of the crack tip.

The presence of a region in which simultaneously high stresses (exceeding the critical *σ*_22_ > *σ*_C_; 1.5 < *η <* 2.4)) and relatively low plastic strain (0.01 < *ε*_eff_plast_ < 0.2) occurred caused realization of a fracture according to the cleavage mechanism. Completely brittle fracture was realized if this region was located immediately before the crack front, and was observed in the lower plateau of the brittle-to-ductile cracking relationship.

The occurrence of high strain levels (0.2 < *ε*_eff_plast_ < 1.6) together with stresses corresponding to a stress triaxiality coefficient of 0.7 < *η* < 1.5 in a region before crack tip led to the mechanism of ductile crack propagation by nucleation of voids on carbide particles or inclusions (leading to void growth and coalescence). Completely ductile fracture by void growth occurred in the higher plateau of the brittle-to-ductile fracture relationship.

Mixed ductile and brittle fracture corresponds to the transition interval of the brittle-to-ductile fracture relationship. In this case, in a region immediately before pre-crack tip, conditions were realized for the ductile crack growth mechanism. Simultaneously, in a region a certain distance from the pre-crack tip, the conditions for brittle fracture were fulfilled.

The results presented in this article conceptually confirm the assumptions of the basic criteria, which were formulated for the ductile fracture mechanism [15,17] and for cleavage fracture [18]. In addition, our results made it possible to conceptualize the levels of the characteristic values of stress and strain necessary to realize the appropriate cracking mechanisms for S355 ferritic steel.

## Figures and Tables

**Figure 1 materials-12-03959-f001:**
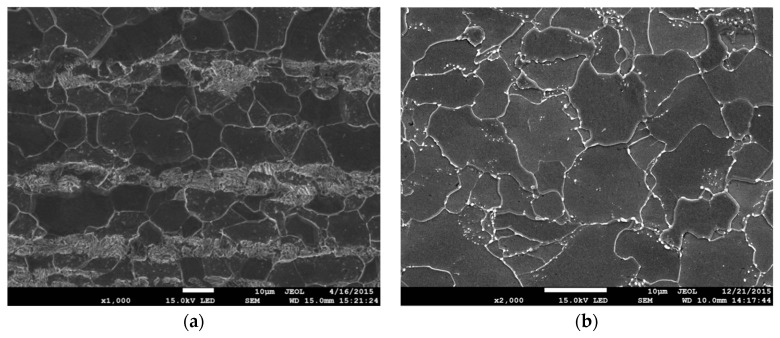
Microstructure of S355 steel. (**a**) Original state and (**b**) after heat treatment.

**Figure 2 materials-12-03959-f002:**
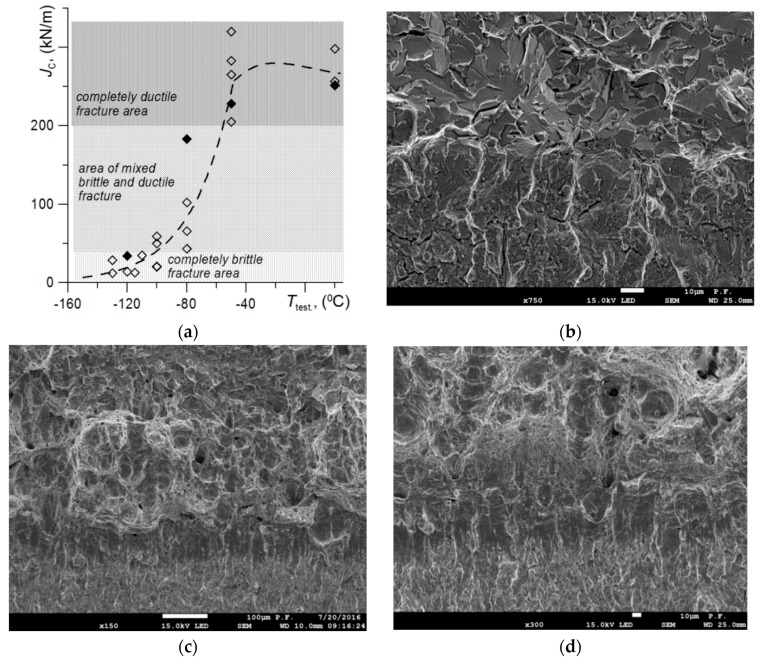
(**a**) Dependence of the *J*_C_ critical values on temperature; (**b**) full brittle fracture at *T* = −120 °C; (**c**) ductile and brittle fracture at *T* = −80 °C and (**d**) full ductile crack growth at *T* = 20 °C.

**Figure 3 materials-12-03959-f003:**
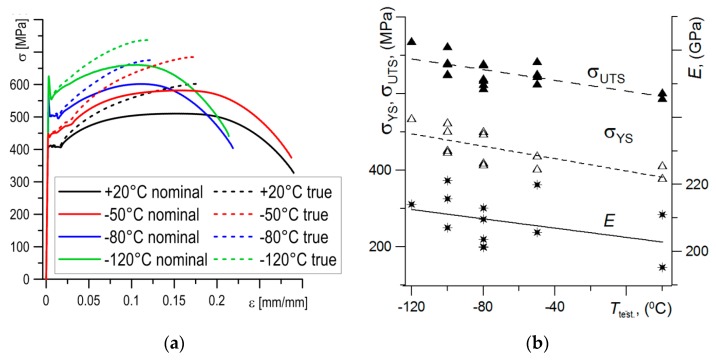
Uniaxial tensile tests results. (**a**) The nominal and true stress–strain dependences for different temperature and (**b**) change of strength properties with temperature.

**Figure 4 materials-12-03959-f004:**
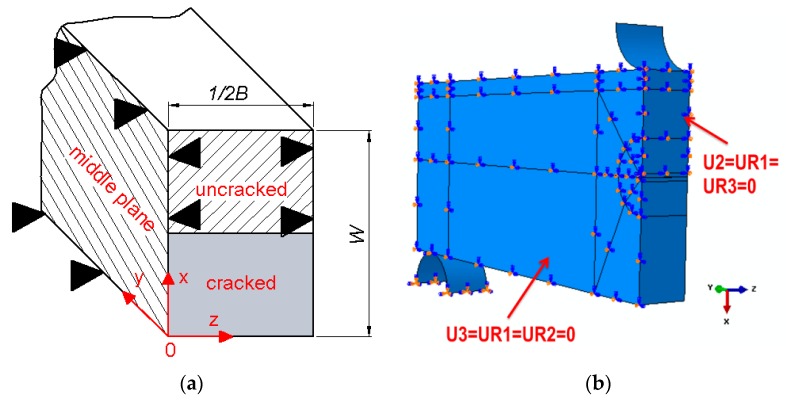
Boundary conditions used in the numerical model of the single-edge notched bend (SENB) specimen. (**a**) Scheme and (**b**) numerical model generated by the ABAQUS program.

**Figure 5 materials-12-03959-f005:**
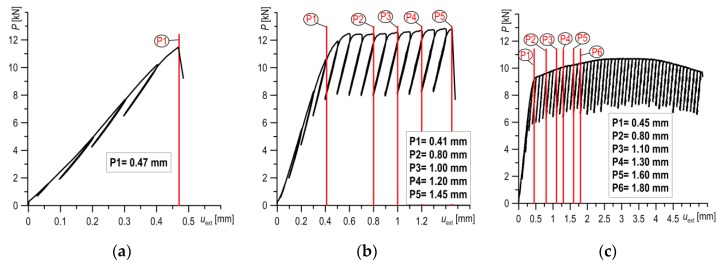
The load–displacement diagrams and load stages (Pi) for specimens tested at different temperatures: (**a**) *T* = −120 °C; (**b**) *T* = −80 °C; and (**c**) *T* = +20 °C.

**Figure 6 materials-12-03959-f006:**
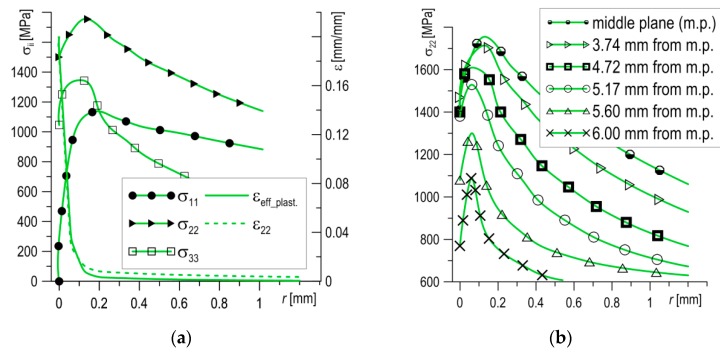
(**a**) The stress and strain distributions in the middle plane of the specimen; (**b**) the stress *σ*_22_ distributions in planes parallel to middle plane; (**c**) the stress *σ*_22_ distributions in the thickness direction; and (**d**) the fracture surface of the specimen at *T* = −120 °C.

**Figure 7 materials-12-03959-f007:**
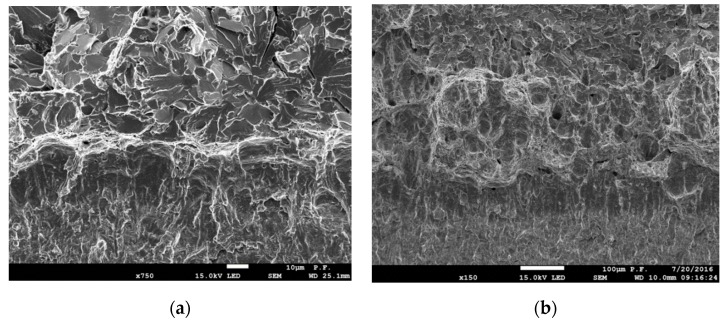
Mixed ductile-cleavage type of fracture. (**a**) At *T* = −100 °C and (**b**) at *T* = −80 °C.

**Figure 8 materials-12-03959-f008:**
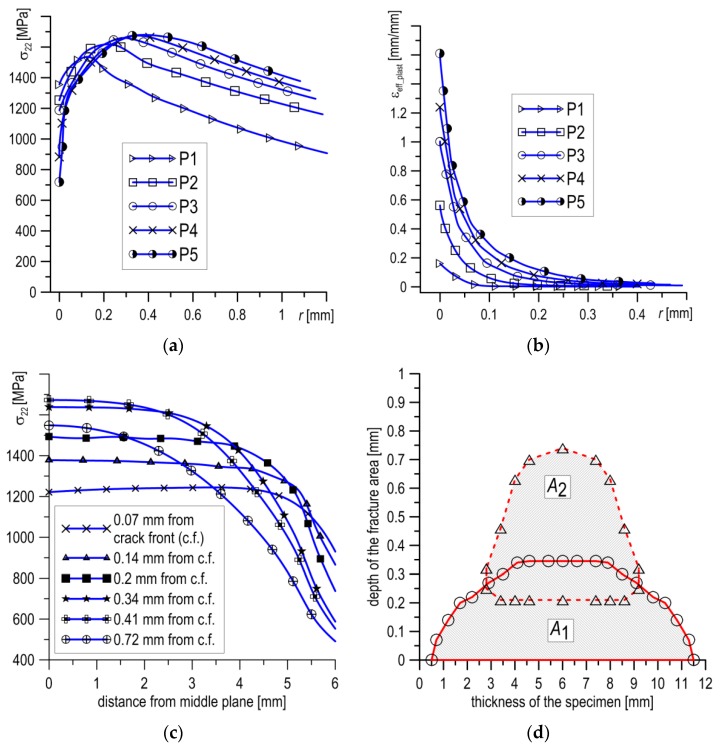
The stress *σ*_22_ (**a**) and strain *ε*_eff_plast_ (**b**) distributions in the middle plane at loading moment Pi. (**c**) The stress *σ*_22_ distributions in the thickness direction and (**d**) the location and dimensions of the critical areas *A*_1_ at *T* = −120 °C and *A*_2_ at *T* = −80 °C.

**Figure 9 materials-12-03959-f009:**
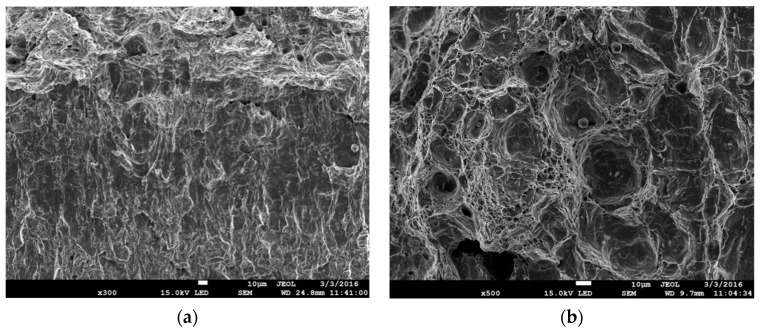
Fully ductile crack growth at *T*_test_ = 20 °C. (**a**) view of stretch zone and initial area of ductile crack growth and (**b**) the ductile mechanism of crack growth through growth of voids.

**Figure 10 materials-12-03959-f010:**
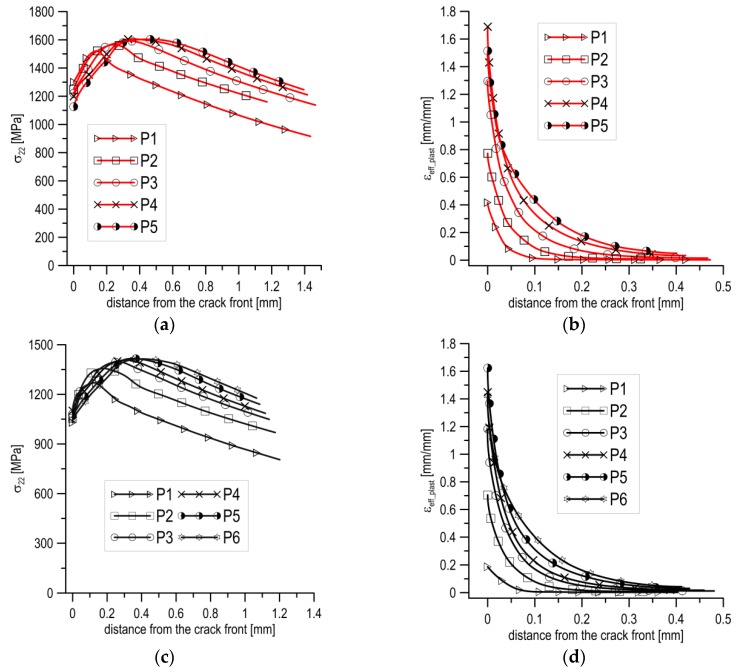
The stress and strain distributions: (**a,b**) – at *T* = −50 °C; and (**c,d**) – at *T* = 20 °C.

**Figure 11 materials-12-03959-f011:**
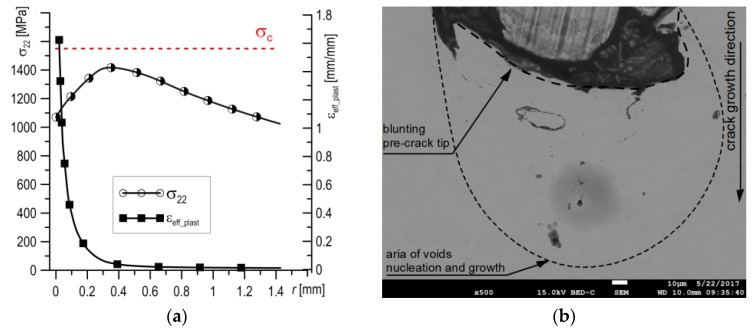
(**a**) Typical stress and strain distributions for the fully ductile crack growth mechanism; (**b**) the area before the blunted pre-crack tip at the start of a ductile fracture; and (**c**,**d**) the voids nucleate around particles of carbide precipitates, located mainly along grain boundaries.

**Figure 12 materials-12-03959-f012:**
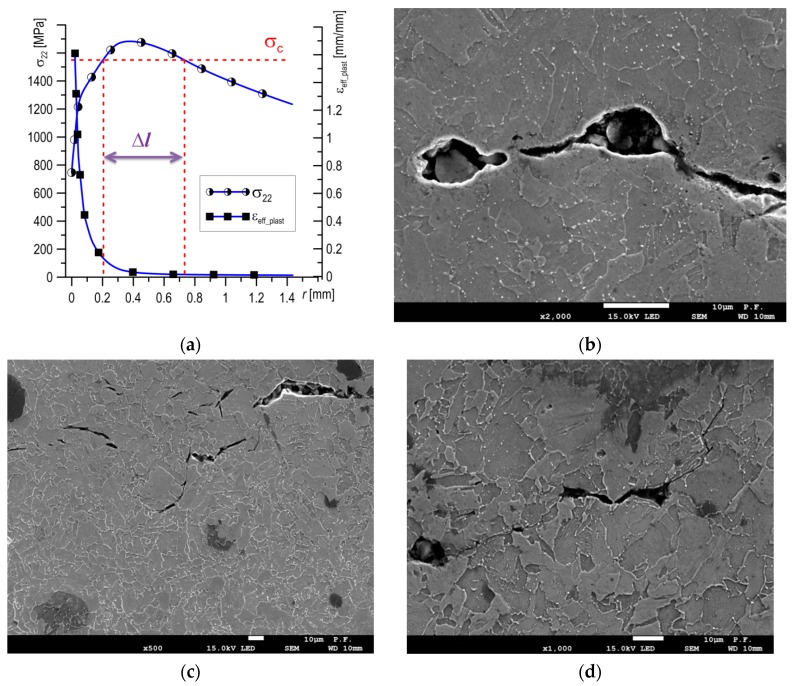
(**a**) Stress and strain distributions and localization zones of ductile and cleavage fractures for the mixed mechanism of crack growth; (**b**) ductile crack propagation by void growth and coalescence; (**c**) cleavage cracking at initiation; and (**d**) a main crack, formed by cleavage.

**Figure 13 materials-12-03959-f013:**
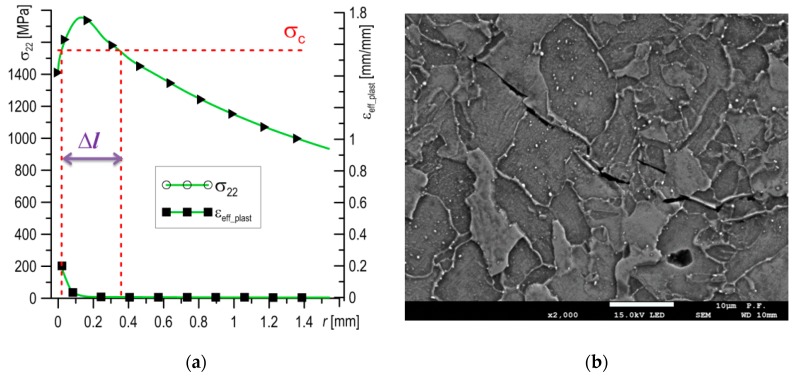
(**a**) Stress and strain distributions for the full cleavage fracture and (**b**) micro-cracks at initiation of the cleavage fracture.

**Table 1 materials-12-03959-t001:** The strength, plasticity and fracture toughness characteristics of S355 steel.

*T* (°C)	*E* (GPa)		σ_YS_L_/*σ*_YS_H_ (MPa)		*σ*_UTS_ (MPa)		*n*		*A*_5_ (%)	*J*_IC_ (kN/m)
	nom	true	nom	true	nom	true	nom	true		
20	194	195	405/412	412/414	510	602	8.13	4.51	28.97	251
−50	204	205	489/498	527/529	578	651	8.98	5.04	28.71	218
−80	206	207	494/558	502/559	601	675	9.36	5.28	21.86	194
−120	213	214	554/624	556/627	661	737	9.85	5.57	21.37	43

where: σ_YS_L_—lower yield strength, σ_YS_H_—higher yield strength, σ_UTS_—ultimate strength, *E—*Young’s modulus, *n*—power exponent.

**Table 2 materials-12-03959-t002:** Some characteristic values of the fracture process of S355 steel.

T (°C)	Cleavage Propagation									Ductile Growth		
	σ_22_max_ (MPa)	$\frac{\sigma_{22\_max}}{\sigma_{YS}}$	$\frac{\sigma_{22\_max}}{\sigma_{c}}$	∆l, (µm) From–To	∆l (µm)	A (mm^2^)	ε_eff_plast_ for ∆l	η for ∆l		ε_eff_plast_ Before ∆l	η	
−120	1760	3.2	1.14 > 1	0–350	350	2.88	< 0.2	1.5–2.4	YES	absent	absent	NO
−80	1676	3.3	1.08 > 1	210–760	550	2.78	< 0.2	1.5–2.4	YES	0.4–1.6	0.7–1.1	YES
−50	1600	3.3	1.03 > 1	270–670	400	1.76	< 0.2	1.5–2.4	NO	0.4–1.6	0.7–1.1	YES
20	1400	3.3	0.90 < 1	absent	absent	absent	absent	absent	NO	0.4–1.6	0.7–1.1	YES

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
