# Peer review of "Fracture Mechanisms of S355 Steel—Experimental Research, FEM Simulation and SEM Observation"

_materials, 2019, doi:10.3390/ma12233959_

Round 1

Reviewer 1 Report

The paper investigated the fracture mechanisms of S355 steel by using experimental  research and FEM simulation. After the reviewing, I think that the paper should be accepted after major revision. The comments are as following:

1.The abstract should clearly describe the summary of this paper.

2. The conclusion should modify. The descriptions are  very roughly and can't understand what is the important result in this manuscript.

3.The S355 steel is very common structural but it also need to present the contents of this steel. It can be identified the specimens under testing.

4. The scale of SENB testing specimen should be given.

5. The author said the Fig. 11b is the view of area before blunted pre-crack tip at start of ductile fracture. I can not understand where is the pre-crack tip. The Figure should give some marks.

6. The Fig. 11c and 11d. the author said the voids nucleated around particles of carbide precipitates located mainly along grain boundaries. It should give what type or content of these precipitates. Because it may be the impurity. 

Author Response

1

Dear Reviewer, thank You very much for review our paper.

According to You suggestion English language is corrected in the paper.

In the abstract introduce more clearly summary of the results presented in paper. The chapter “Conclusions” has been rewritten. The chemical composition of S355 steel is introduce into text (line 100-102). The dimensions of the SENB specimens added in text (line 127). The Fig. 11b has been completed. The blunted pre-crack and aria of voids initiation are marked. In 11c and 11d You can see microstructure of tested steel: the grains of ferrite with particles of carbide precipitates, which located along grain boundaries. In current experiments we have not conducted research to identify types of these particles. However, from our earlier research, we believe that these may be e.g. (M3C; M23C6; M7C3)-type coagulated carbide particles.

Reviewer 2 Report

It is a good paper for studying the fracture characteristics of S355 steel depending on the temperature. However, the detailed description for analysis and the introduction of equations for the level of plastic deformation.

The clarity of Figure 2. (a) needs improvement. Three variables in Table 1. are missing for understanding.

The main caption for Fig 3. is necessary.  The boundary condition for numerical anlaysis in Fig. 4 should be explained.

Typo error is in the title of 5. (discussion). The consistency of font sizes in subscripts for variables are kept.

The setences in Chap 5. needs rewriting.

Author Response

Dear Reviewer, thank You very much for review our paper.

According to You suggestion English language is corrected in the paper.

The variables in the Table 1 have been explained. The caption to Fig. 3 is corrected. Also, introduced Abaqus generated scheme of boundary conditions and some explanations into text. Title of Chapter 5 has been changed on "Discussion" (technical error). The sentences in Chapter 5 have been rewritten.

Round 2

Reviewer 1 Report

All comments have modified. It can be published in this form.

Author Response

Dear Reviewers,

Thank you for the comments given in the review.